# Growth of Nanostructured TiO$_2$ Thin Films onto Lignocellulosic Fibers through Reactive DC Magnetron Sputtering: A XRD and SEM Study

**Telmo Eleutério** [1], **Susana Sério** [2] and **Helena C. Vasconcelos** [1,2,*]

[1] Faculdade de Ciências e Tecnologia, Universidade dos Açores, 9501-801 Ponta Delgada, Portugal; telmo.mf.eleuterio@uac.pt
[2] Laboratory of Instrumentation, Biomedical Engineering and Radiation Physics (LIBPhys-UNL), Department of Physics, NOVA School of Science and Technology, NOVA University Lisbon, 2829-516 Almada, Portugal
[*] Correspondence: helena.cs.vasconcelos@uac.pt

**Abstract:** TiO$_2$ thin films were deposited on ginger lily (*Hedychium gardnerianum*) fibers using a custom-made DC reactive magnetron sputtering system with Ar/O$_2$ mixture at two O$_2$/(O$_2$ + Ar) ratios (50% O$_2$ and 75% O$_2$) and sputtering powers (500 and 1000 W), and their effects on the structure and surface morphology of TiO$_2$ films were investigated. XRD analysis showed the presence of the mainly anatase phase in the deposited films, with a small fraction of rutile phase detected for TiO$_2$ deposited with the higher oxygen percentage and sputtering power. SEM imaging revealed that the films exhibited distinct surface features depending on the deposition conditions. Specifically, films deposited with 50 O$_2$ % and 1000 W exhibited porosity, while the films deposited under other conditions appeared either dense with a cauliflower-like appearance or displayed surface features resembling lines and mountain ranges of coalesced particles. The grain size of dense films increased with increasing sputtering power. The deposition conditions significantly affected the resulting surface topography, with an increase in surface roughness parameters observed for both power levels when the oxygen concentration in the deposition atmosphere was increased from 50% to 75%. The adhesion tests conducted using sonication and EDS analysis revealed that almost all of the studied films exhibited good adhesion, as evidenced by the atomic content (at. %) of Ti remaining intact after sonication, indicating good adhesion. However, the porous film exhibited a slightly lower adhesion grade, suggesting that the porous structure may have influenced the adhesion properties.

**Keywords:** titanium dioxide; porous films; magnetron sputtering; plant fibers; *Hedychium gardnerianum*

## 1. Introduction

The growing awareness of the negative impact of synthetic fibers and materials on the environment has led to a greater focus on using natural fibers in various industries [1,2]. The use of invasive species such as ginger lily (*Hedychium gardnerianum*) as a source of cellulosic fibers [3,4] can have a positive impact on the environment as it helps to control their spread and reduce their impact on native ecosystems [5]. Alkaline pre-treatment with NaOH is an effective way to remove impurities [6] and obtain pure cellulose fibers, which can be used in a variety of applications, including the production of biodegradable materials, composite materials, and textiles. In addition, these fibers are also renewable, which makes them a more environmentally responsible choice compared to synthetic fibers that can take hundreds of years to decompose.

However, the plant fibers have very complex hierarchical structures, consisting of several bundles of fibrils linked to each other by aromatic compounds and other biomolecules of high molecular weight, such as hemicellulose and lignin [6]. Although such structures are perfectly adapted to the plant physiology, after extraction, the natural surfaces are often

not ideal for technological applications, especially because of their hydrophilic characteristic due to the large amounts of hydroxyl groups, –OH, in the cellulose structure [7,8]. Nevertheless, −OH groups are common sites for chemical functionalization because of their reactivity [9]. This allows for the introduction of new functionalities and the customization of the chemical and physical properties of the fiber for a wide range of applications, including improving the mechanical and thermal properties of fibers, as well as new properties, such as antimicrobial activity, hydrophobicity, and photocatalytic activity.

$TiO_2$, or titanium dioxide, is a widely used material in various industries, such as photocatalysis, solar cells, self-cleaning surfaces, and antibacterial coatings [10], due to its unique optical and electrical properties. The growth of $TiO_2$ films on unheated substrates has been an area of interest for researchers as it opens up the possibility of producing low-cost and scalable coatings. The $TiO_2$ crystal structure can affect its properties, with anatase being more active than rutile in terms of photocatalytic activity [11]. However, even amorphous $TiO_2$ nanoparticles have demonstrated photocatalytic and antibacterial properties, although to a lesser extent [12].

Cotton fabrics functionalized with $TiO_2$ particles through a dip-pad-dry-cure [13] or sol-gel methods [14] can result in photoactive films on the fabric surface. Despite this, these coatings may not always be resistant to abrasion and washing [15], which can lead to a loss of functionality over time. The release of $TiO_2$ particles into the environment can also be a concern. Therefore, the durability and stability of functional coatings are important factors in the design of functional fibers, and the choice of deposition method can play an important role in determining the performance of the functional coating.

The Reactive DC Magnetron Sputtering (RDCMS) is a popular technique for depositing $TiO_2$ thin films with controlled composition and stoichiometry [16], by combining sputtering with reactive gas chemistry. This process allows the obtaining of high-quality and uniform coatings with high adherence to the substrate [17], which can improve the durability and stability of the functional coatings. RDCMS is indeed an environmentally friendly process, and it has been used to deposit various functional materials on different substrates. Moreover, the film structure and properties can be controlled by the process parameters, such as argon/oxygen flow rate, sputtering power, pressure, and target composition. In RDCMS, argon is typically used as a sputtering gas to create a plasma and generate ions and atoms, and the oxygen is added to the process to react with the deposited material and control the film composition and stoichiometry. By adjusting the gas flow rates and partial pressure during sputtering, it is possible to control the composition of the deposited film. In the case of $TiO_2$ films, the interplay between argon and oxygen gases is crucial for regulating the film's structure [18]. Argon serves as the sputtering gas, generating ions and atoms while controlling the particle flux onto the substrate. Meanwhile, oxygen governs the film's composition and stoichiometry, influencing surface reactions and the mobility of reactive species that affect the film's structure.

Similarly, in the case of Ti-Cr-N coatings, the nitrogen content in the film can be controlled by modifying the $Ar/N_2$ gas flow ratio, which has a direct impact on the coatings' microstructure and surface morphology. Changing the gas flow ratio also alters the surface morphology of the coatings. For instance, lower $Ar/N_2$ ratios lead to a pyramidal structure with higher surface roughness, while higher $Ar/N_2$ ratios result in a smoother, cauliflower-like structure with lower surface roughness [19].

In another study that investigated the effects of oxygen content and gas pressure on the structural properties of ZnO thin films deposited by DC magnetron sputtering [20], it was found that low oxygen contents led to the formation of porous structures, while higher oxygen contents resulted in denser films.

Additionally, the DC sputtering power and the plasma gas composition (argon/oxygen ratio) are two critical parameters that have been shown to have a significant effect on the structural and morphological properties of $TiO_2$ films. The DC sputtering power determines the energetic conditions under which the atoms are deposited onto the substrate and affects the size, shape, and orientation of the deposited grains. The plasma gas composition, on

the other hand, influences the stoichiometry of the film and the availability of oxygen to participate in the formation of the crystalline structure [18]. It is important to note that the interplay between the various process parameters can be complex and the optimal conditions for a particular film may require careful experimentation and optimization. Moreover, the process parameters can have a significant impact on the surface morphology of the $TiO_2$ film deposited by reactive DC magnetron sputtering [18]. By carefully tuning these parameters, the desired properties in the deposited $TiO_2$ film can be achieved, such as a uniform thickness, specific crystalline phase, and a high level of surface roughness or smoothness. Moreover, the growth of anatase $TiO_2$ films on unheated substrates through sputtering [21] is a promising approach for producing high-quality $TiO_2$ coatings for various applications. The roughness of the substrate can also determine the growth mode of the $TiO_2$ film. Usually, the surface roughness is beneficial for nucleation formation and grain growth, resulting in greater grain size in deposited films [22]. On the other hand, a longer deposition time can lead to larger grain sizes due to the increased diffusion of atoms on the film surface. However, this relationship is not linear and depends on various factors, such as the substrate material and film composition. Nevertheless, the main factor to tailor the crystallization of $TiO_2$ sputtered film, remains undetermined.

Most of the studies found on the literature about $TiO_2$ deposition by RDCMS were carried out on smooth substrates, mainly in microscope slides, Si-wafers [18,23], or in synthetic fibers [24], whose surfaces are equally smooth, although curved. Only a few studies report the use of naturally rough substrates or natural fibers. The smooth substrates allow for more uniform and controlled deposition of the film, which is important for obtaining reproducible results in research studies. On the other hand, naturally rough substrates and plant fibers have irregular and complex surface morphologies, which can affect the uniformity and control of the deposited film, leading to more variability in the results. Despite these challenges, there has been a growing interest in using naturally rough substrates and natural fibers in sputtering studies, as they have potential applications in many fields where the natural texture and morphology of the substrate can play a critical role in the performance of the deposited films.

In our previous publication, we examined by XPS and FTIR the characteristics of sputtered $TiO_2$ films on ginger lily fibers [4]. This work aims to understand the factors that influence the crystalline structure, morphology, and adhesion of the $TiO_2$ films to ginger lily fibers. The XRD analysis was used to determine the crystalline structure of the films, while the SEM/EDX analysis was performed to evaluate the surface morphology and chemical composition of the films. Tests were also carried out to evaluate the adhesion of the films to the ginger lily fibers. The results showed that the sputtering conditions, such as the $Ar/O_2$ mixture and sputtering power, can be used to control the properties of the $TiO_2$ films. This information can help to optimize the deposition process and develop new applications for these films.

## 2. Materials and Methods

### 2.1. Fibers

The ginger lily fibers were obtained from plants grown in S. Miguel Island, Azores (Portugal). The stems were treated with a 5 wt. % NaOH solution at 80 °C for 30 min to remove any micro-organisms, non-cellulosic compounds, and polysaccharides. The fibers were then washed with water to remove any remaining traces of alkali and dried in a greenhouse. The purpose of the NaOH treatment was to remove hemicelluloses and other impurities, as indicated by previous FTIR observations [3].

### 2.2. Thin Film Deposition

The magnetron sputtering was realized in a custom-made DC magnetron sputtering system, using as a target a titanium (Ti) disc (99.99% of purity, Goodfellow, Cambridge, UK). In the preparation of titanium dioxide films, argon (Air Liquid, Paris, France, 99.99%) was also used as the sputtering gas and oxygen (Gás Piedense gases, Setúbal, Portugal, 99.99%)

was used as a reaction gas. To achieve a base pressure of $10^{-4}$–$10^{-5}$ Pa (before introducing the sputtering gas) a turbomolecular pump (Pfeiffer TMH 1001, Pfeiffer Vacuum GmbH, Asslar, Germany) was utilized. Before the sputter-deposition of the films, a movable shutter was placed between the target and the supports. The target was pre-sputtered in an Ar atmosphere for 1 min to remove the target surface oxidation. All obtained films were grown at room temperature. No additional heating was applied to avoid fiber damage. $TiO_2$ film deposition was performed at a constant pressure, $p_T$ = 2.2 Pa, combining two different oxygen ratios (50% and 75% $O_2$) with two different sputtering powers ($W_p$ = 500 and $W_p$ = 1000 W), respectively, through deposition times of $t_d$ = 50 min and $t_d$ = 75 min. The target substrate distance, d = 100 mm, was kept constant. The substrate temperature was measured by a thermocouple passing through a small hole in a copper piece, which was placed in contact with the substrate and revealed an increase from 60 to 100 °C, after sputtering power change from 500 to 1000 W. Other details about the sputtering experimental conditions have been previously published in previous studies [3,18].

### 2.3. Fibers and TiO$_2$ Thin Films Characterization

The crystalline structure of the fiber and of the films was determined by X-ray diffraction (XRD) using a D8 Advance Bruker AXS θ–2θ diffractometer (Bruker AXS GmbH, Karlsruhe, Germany) and $Cu_\alpha$ radiation with a wavelength of λ = 0.15406 nm. Diffraction patterns were recorded for 2θ from 10° to 27° for untreated fibers, and 2θ from 22° to 50° for TiO$_2$-coated fibers, with a scan rate of 0.03° min$^{-1}$.

The samples surface morphology was analyzed by Scanning Electron Microscopy (SEM) using a Hitachi S-2400 (Hitachi, Krefeld, Germany) with thermionic (W) emission gun (Hitachi, Krefeld, Germany), resolution of 5 nm at 25 kV, equipped with secondary and backscattered electron detectors and digital image acquisition. The chemical composition of the films was evaluated by SEM/EDS with a Bruker Quantax energy dispersive X-ray spectrometer (EDS) (Bruker, Billerica, MA, USA) and light elements detector (Bruker, Billerica, MA, USA).

Surface analysis has been conducted using the Mountains®9 software (Trial version), developed by Digital Surf in Besançon, France (https://www.digitalsurf.com/, accessed on 23 November 2022).

### 2.4. Evaluation of TiO$_2$ Thin Film- Fiber Adherence

The adherence test of the as-deposited films was performed by a sonication treatment for 60 min in an equipment UIS250V of 250 W (Hielscher Ultrasonics GmbH, Teltow, Germany) in a continuous mode at 70% of amplitude of the sound waves. The amount of TiO$_2$ was checked before and after the sonication process using scanning electron microscopy with energy-dispersive X-ray spectroscopy (SEM/EDS).

## 3. Results and Discussion

### 3.1. Fiber's Morphology, Structure, and Composition

In Figure 1, a representative SEM/EDS spectra of ginger lily fibers is depicted. The analysis of the SEM micrographs reveal that the surface of ginger lily fibers is not smooth and uniform. As can be observed, the extracted fibers have a multi-fibrillar structure (Figure 1a), and their surface is rough, with a wavy texture, wrinkles, grooves, and smooth areas (Figure 1b). Although efforts have been made to remove non-cellulose substances, such as pectin and wax, some organic residues (e.g., xylem) still remain on the fiber surface, as shown in Figure 1c. The EDS chemical analysis of untreated ginger lily fibers revealed that the main elements on the surface are carbon (C) and oxygen (O), as expected and as observed in Figure 1d. The ratio of oxygen to carbon (O/C ratio) is typically 0.83 for cellulose when hemicellulose and pectin are present [25]. However, in this study, the O/C ratio was found to be 0.85, a value slightly higher, indicating that most of the waxes and lignin have been removed from the fiber surface [26]. This leads to an increased fraction of cellulose and more hydroxyl groups (C–OH bonds) on the fiber surface, as revealed by XPS

chemical analysis (Figure 2a), and it has been reported that these hydroxyl groups can react with various functional groups [4,27].

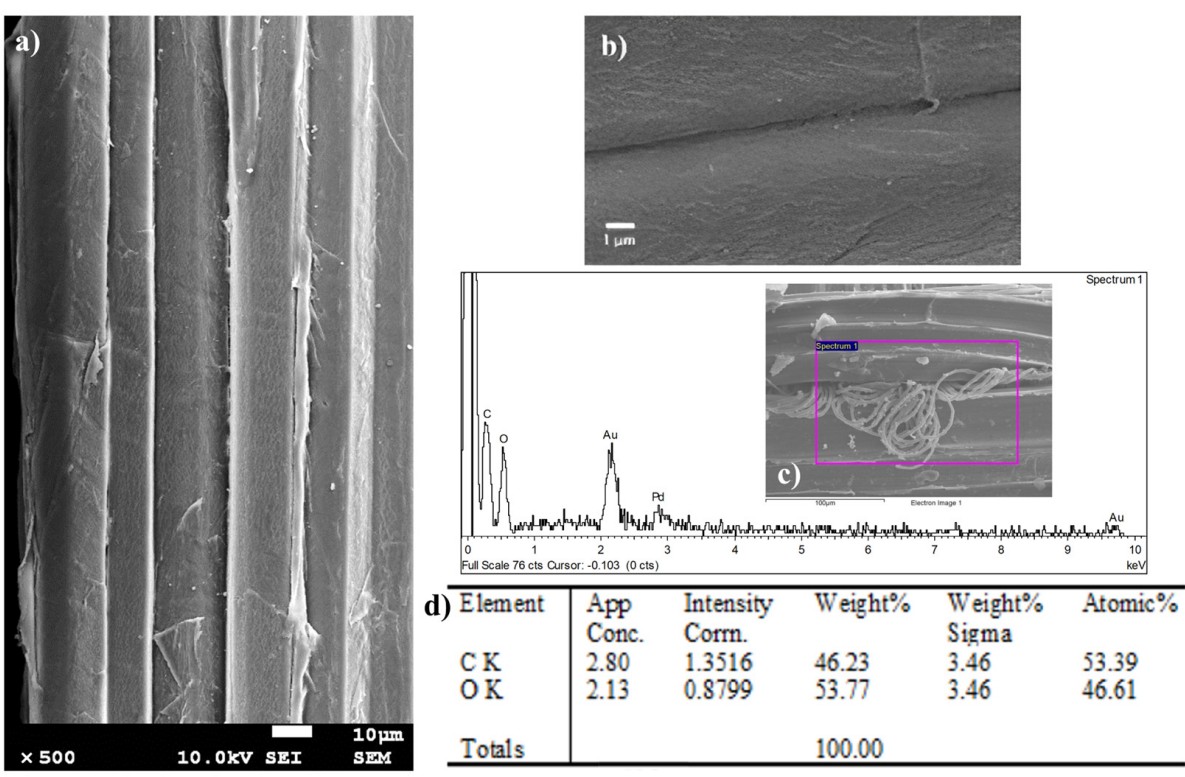

**Figure 1.** Representative SEM/EDS spectra of ginger lily fibers: (**a**) Typical multi-fibrillar structure; (**b**) detail of the appearance of the surface; (**c**) traces of xylem; and (**d**) Typical EDS spectrum showing how characteristic X-rays correspond to the principal fiber elements (Au and Pd are from conductive film and sample holder, respectively).

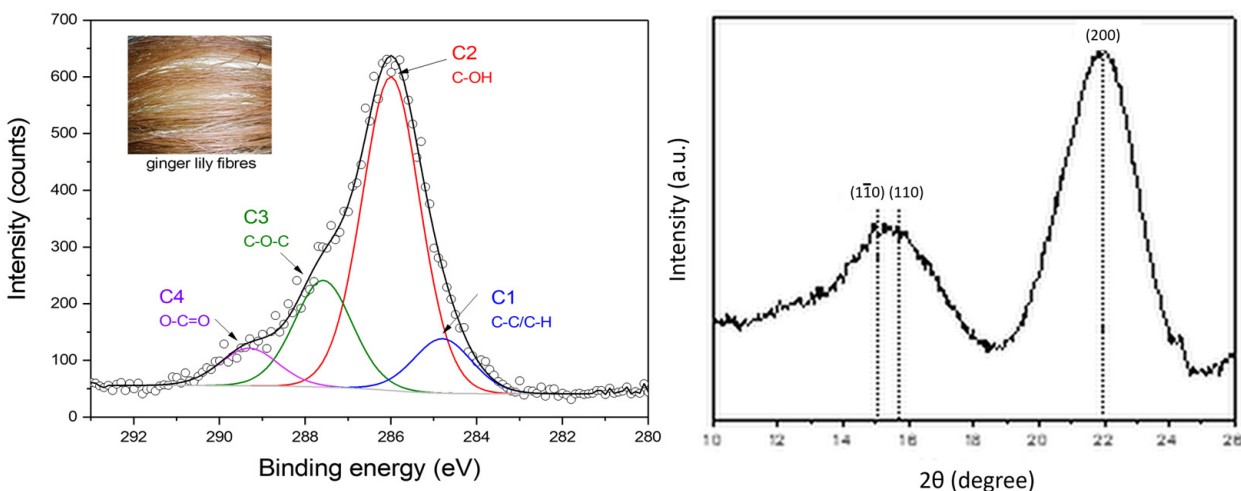

**Figure 2.** (**a**) C1s XPS spectra of a ginger lily surface; (**b**) XRD pattern of a ginger lily fiber.

The structure of ginger lily fibers consists of a combination of crystalline and amorphous microfibrils [3]. This structure determines the reactivity of the fibers. The XRD pattern of ginger lily fibers can be seen in Figure 2b. The observed two large diffraction peaks are commonly found in most plant fibers. The diffraction peaks at 2θ around 15.1°, 15.6°, and 21.9° were identified by JCPDS-ICDD:50-2241 and are attributed to the planes

($1\bar{1}0$), (110) and (200), respectively, which are consistent with the characteristic diffraction peaks of cellulose [28]. Moreover, this is in good agreement with the results of the EDS analysis.

The crystallinity Index of ginger lily fibers is 69.77% [3] and there are also present amorphous constituents, as indicated by a peak at $2\theta = 15.6°$. To note that the more crystalline the fibers are, the more rigid they become and less flexible they are [29]. The (200) line is an indicator of the crystalline structure of the fibers, and a decrease in its intensity usually means that the structure has been degraded. In this case, however, the intensity of the (200) line is not low, indicating that the crystalline structure has not been degraded.

### 3.2. Structure of the Fibers after TiO₂ Deposition

The stability of the crystalline structure of the fibers was evaluated using XRD analysis after the deposition of the TiO₂ coatings. The results, presented in Figure 2, show that the untreated fibers have the expected crystalline structure of cellulose. However, when the TiO₂ films were deposited, the cellulose structure changed as can be noted in Figure 3 between 20–22° (2θ degrees) and 10°–20° (2θ degrees). These changes suggest that there may have been modifications in the crystalline structure of the cellulose, although its composition remains the same. The decrease in the (200) peak is an indicator of the presence of disorder in the cellulose. Additionally, the broad peak at $2\theta = 15.6°$, which is related to the amorphous structure, becomes more prominent after the TiO₂ deposition, and shifts to lower degrees, consistent with an amorphous cellulose structure [30].

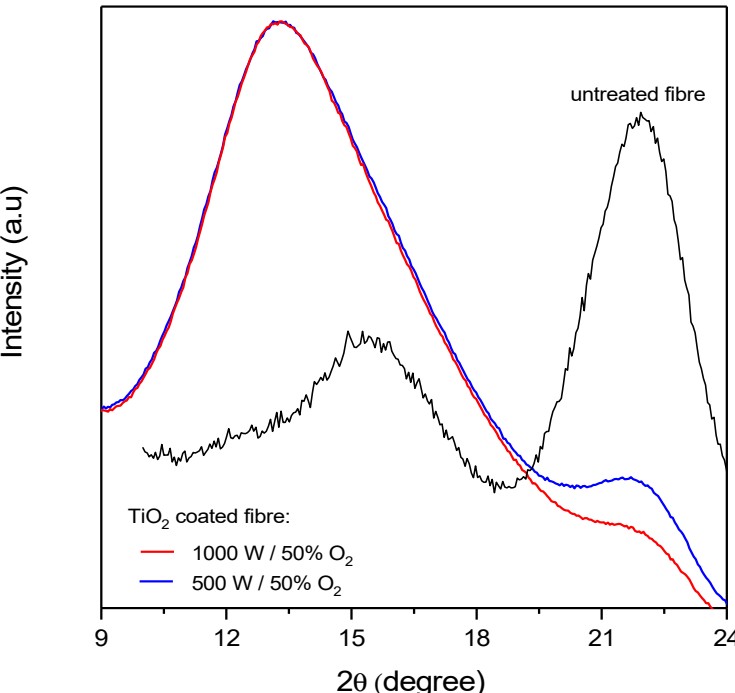

**Figure 3.** X-ray diffraction patterns of untreated fibers and TiO₂-coated fibers produced at two different sputtering powers and with 50% O₂.

This means that plasma oxygen particles can easily cause oxidation in the crystalline regions of cellulose, which results in amorphous cellulose. Our findings suggest that plasma oxygen particles can readily initiate oxidation in the crystalline regions of cellulose, leading to the formation of amorphous cellulose. This oxidation phenomenon was previously observed in our research [4], where we identified an increase in the presence of C3 and C4 contributions in the XPS spectrum. These contributions are associated with O–C–O and COO– groups.

### 3.3. Growth of TiO$_2$ Films in Different DC Sputtering Conditions

The cellulose fibers acquire a negative charge after being exposed to the reactive sputtering because of the presence of carboxyl and hydroxyl groups [31]. After extraction, namely treatment with NaOH, these surface groups (COO–) become more accessible, which results in an increase in the fibers' surface free energy [32]. This, in turn, improves the interaction with charged particles during sputtering compared to the interaction with neutral particles. The decrease in the fiber crystallinity degree also contributes to this improvement. Moreover, the process of film growth can be assumed as a self-assembly of nanometer-sized clusters that are charged and formed in the gas phase, known as the charged cluster model. These charged clusters interact with the substrate surface upon impact during sputtering. In other words, the formation of the films is thought to occur through the aggregation of tiny, charged particles in the gas phase, which then come into contact with the substrate surface and adhere to it, eventually building up into a complete film [33]. The model considers various physical and chemical parameters that can influence the growth of films by DC reactive sputtering, including DC power, the process pressure, gas composition, and substrate temperature.

However, the relative importance of these different parameters may vary with respect to the structure of the film. For example, some parameters may be more effective in promoting the growth of crystalline structures in the film, while others may be less effective. The conditions that favor porous film deposited on plant fibers by DC reactive sputtering can be controlled by adjusting the argon/oxygen ratio, sputtering power, and pressure.

### 3.4. Growth of TiO$_2$ Films in Different DC Sputtering Conditions

The crystalline structure of TiO$_2$ films deposited on the surface of ginger lily fibers was analyzed using XRD. The XRD patterns of the TiO$_2$ films produced at different sputtering powers (500 and 1000 W) and oxygen concentrations (50% and 75%) are presented in Figure 4. The results show that films produced at 500 W are amorphous, while those produced at 1000 W have a polycrystalline structure with diffraction peaks consistent with anatase TiO$_2$. These peaks are primarily oriented at (101), (004), and (200) with 2θ values of 25°, 37° and 47°, respectively, which is typical of tetragonal anatase TiO$_2$ phase (JCPDS 21-1272).

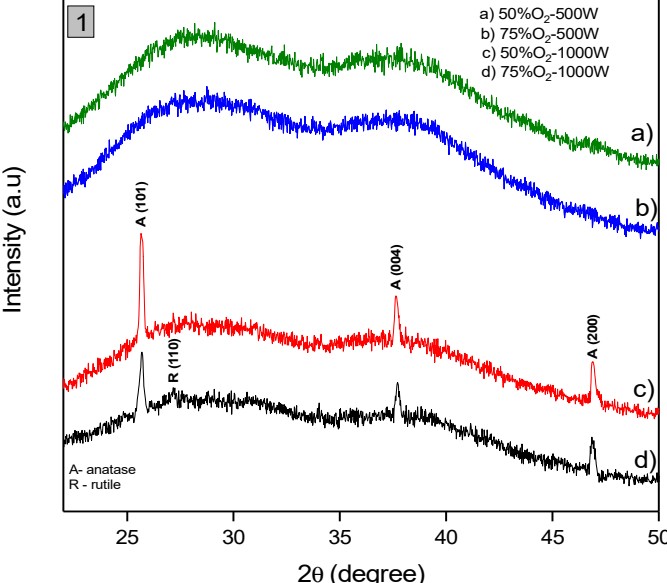

**Figure 4.** *Cont.*

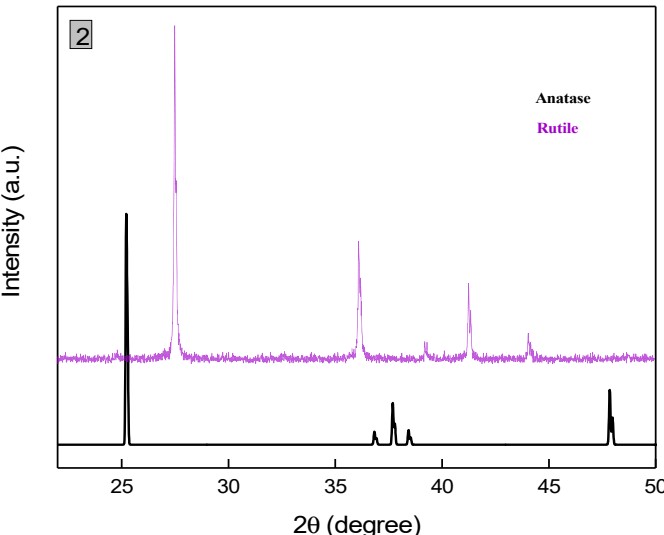

**Figure 4.** X-ray diffraction patterns of TiO$_2$ coated fibers under different sputtering powers (500 W and 1000 W) and varying O$_2$ deposition conditions (50% O$_2$ and 75% O$_2$) in Graph (**1**). Graph (**2**) presents XRD patterns of Anatase and Rutile standards for comparison.

The results of the XRD analysis also demonstrate that the TiO$_2$ films deposited with 50% O$_2$ on ginger lily fibers, anatase was the dominant phase. However, when the oxygen concentration was increased to 75%, one very slight additional peak related to the rutile phase is formed. This peak have orientation (110) for a 2θ value of 27°. The presence of broad and diffuse X-ray diffraction patterns for the films deposited at 1000 W indicate that they are poorly crystallized.

The crystallite sizes for the films was estimated using X-ray analysis and the Scherrer equation [34]:

$$D = 0.94 \times \lambda / \beta \times \cos \theta \tag{1}$$

where λ is the wavelength of CuK$_\alpha$ radiation, β is the corrected full width at half maximum (FWHM) of diffracted peaks and θ the Bragg diffraction angle. The Scherrer equation requires angles to be expressed in radians. The result is expressed as D, which represents the average size of the crystallites in the film. Correction for the line broadening by the instrument was applied as in [18]. The crystallinity ($\chi_c$) or amount of a specific crystalline phase in the films, was determined by the equation:

$$\chi_c = \frac{\text{Area of cristalline peaks}}{\text{Area of all peaks (crystal. + amorph.)}} \times 100 \tag{2}$$

The $\chi_c$ was calculated for the films that were deposited using two different oxygen concentrations (50%O$_2$–1000 W and 75%O$_2$–1000 W). The calculation involved determining the fraction of the anatase phase in the films produced using the 50%O$_2$–1000 W condition and the fraction of both the anatase and rutile phases in films produced using the 75%O$_2$–1000 W condition. The resulting value represents the amount of crystalline phase in the film.

The presence of small, Internal stresses or deformations in the sputtered films was analyzed and, in particular, the Stokes and Wilson formula [35] was used to study and estimated the micro-strain, ε, which was a measure of the internal stress or deformation in the films:

$$\varepsilon = \beta / (4 \times \tan \theta) \tag{3}$$

To obtain micro-strain values using Equation (3), the diffraction angles and the full width at half maximum (FWHM) of the diffraction peak must also be converted from

degrees to radians. The results of the X-ray diffraction analysis, including the crystallinity and micro-strain values, are summarized in Table 1.

**Table 1.** Average crystallite size (D), crystallinity ($\chi_c$), and micro-strain ($\varepsilon$) of the sputtered $TiO_2$ films at 1000 W and deposited with 50% and 75% $O_2$.

| Sample | FWHM/(°) | Peak Position 2θ/(°) (hkl) | Average Crystallite Size (D)/nm | $\bar{D}$/nm | Crystallinity ($\chi_c$)/% | Micro-Strain ($\varepsilon$) Along (hkl) Plane |
|---|---|---|---|---|---|---|
| 50%$O_2$-1000 W | 0.21 | 25.69 (101) | 41 | | | 0.0040 (101) |
| | 0.23 | 37.64 (004) | 38 | 39 | 1.5 | 0.0030 (004) |
| | 0.23 | 46.90 (200) | 39 | | | 0.0024 (200) |
| 75%$O_2$-1000 W | 0.25 | 25.65 (101) | 34 | - | | 0.0049 (101) |
| | 0.27 | 37.68 (004) | 32 | 33 | 0.8 | 0.0035 (004) |
| | 0.28 | 46.80 (200) | 32 | - | | 0.0029 (200) |

Films deposited with 50% oxygen concentration at 1000 W show higher crystallinity ($\chi_c$) and larger average crystallite size (D) compared to films deposited with 75% oxygen concentration at 1000 W. This suggests that higher oxygen concentration may lead to lower crystallinity and smaller crystallite sizes.

The low crystallinity observed in the films produced using the 50%$O_2$–500 W and 75%$O_2$–500 W conditions is thought to be due to the low energy and mobility of the deposited particles, which do not have enough energy to form a crystalline structure. However, the slightly improved crystallinity detected in the films produced using the 50%$O_2$–1000 W and 75%$O_2$–1000 W parameters can be attributed to an increase in particle energy, which allows for better collision and interaction with the fiber surface. This, in turn, improves the conditions for a crystalline phase formation. These findings are supported by the references [36,37].

The increased energy of the particles deposited using 50%$O_2$–1000 W and 75%$O_2$–1000 W parameters can compete with the effects of temperature on crystallization. Normally, the anatase phase forms at around 350 °C, while the rutile phase forms above 600 °C [38]. According to Figure 4, the XRD peaks in the films produced using the 75%$O_2$–1000 W sputtering condition are less intense and wider, which could be due to the small size of the crystals. Conversely, the films produced using the 50%$O_2$–1000 W deposition parameters show increased height and width of the XRD peaks, indicating higher crystallinity and larger crystal size. The final crystalline structure that forms on the substrate depends on the fraction of oxygen in the gas mixture, which influences the partial pressures of both oxygen and argon. As the oxygen content decreases in the argon–oxygen mixture, the crystalline structure of the $TiO_2$ films shifts from a mixture of anatase and rutile to pure anatase. This behavior was also observed by Madaoui et al. [39].

An increase in oxygen during the deposition of $TiO_2$ (75% $O_2$) seems to result in a reduction in the anatase phase intensity and an increase in the rutile phase, although not very significant. This is because rutile is a more thermodynamically stable phase of $TiO_2$ and requires more energy for its formation compared to anatase. With an increase in the oxygen concentration, the target material gradually oxidizes and the Ti atoms are replaced by $TiO_2$ target molecules. As a result, the concentration of Ti atoms in the vapor phase decreases and the nucleation kinetics change [40], leading to a reduction in the crystallite size. This reduction in size is also accompanied by an increase in the micro-strain along certain crystal planes. As crystallite size reduces, the crystallite boundary volume increases,

which leads to an increase in the concentration of lattice imperfections [18]. Moreover, it is detected that when oxygen percentage increases, the micro-strain ($\varepsilon$) along the (101), (004), and (200) planes also increase as shown in Table 1. Similar results have also been obtained previously [18]. Indeed, films deposited with 75% oxygen concentration at 1000 W show slightly higher micro-strain ($\varepsilon$) values along the (101) and (004) planes compared to films deposited with 50% oxygen concentration at 1000 W. This indicates higher internal stress or deformation in the films with higher oxygen concentration.

The type of crystal phase that forms on the substrate is determined by the amount of oxygen and the partial pressures of $pO_2$ and $pAr$. The presence of high oxygen levels (75%) may also impact the crystallization process and lead to a decrease in crystallite size. The energy brought to the surface by the particles (atoms or ions) plays a crucial role in determining the formation of either the anatase or rutile phase, with the anatase phase being formed when the energy is low, and the rutile phase being formed when the energy is high. The anatase and rutile phases are two of the most commonly occurring crystalline structures of $TiO_2$ and are produced through different reaction pathways. The formation of the anatase phase of $TiO_2$ is usually associated with the reaction between metallic Ti and neutral $O_2$ or $O_2-$, whereas the formation of the rutile phase is typically associated with the reaction between $Ti^+$ or activated Ti and $O_2-$ [41]. The specific conditions under which each phase forms, such as temperature, pressure, and oxygen availability, play a crucial role in determining the final crystalline structure.

Furthermore, the species that bring energy to the fiber surface, whether it be condensing atoms or plasma ions, can also play a role in the formation of the anatase or rutile phase, depending on their energy levels and reactivity [42].

Our findings indicate that films deposited with 50% oxygen concentration at 1000 W demonstrate superior crystalline structure in comparison to those deposited with 75% oxygen concentration at 1000 W. This is evident from their higher crystallinity, larger crystallite sizes, and lower micro-strain values, suggesting a more well-ordered and less strained crystalline structure.

*3.5. Microstructure and Surface Morphology of TiO$_2$ Films*

The results of the SEM images depicted in Figure 5 illustrate the surface morphology of thin films obtained by reactive sputtering under different conditions of $O_2$/Ar mixture atmospheres (50% and 75% $O_2$) and deposition powers (500 and 1000 W). Four different microstructures of the films can be attributed to the different deposition conditions used during the reactive sputtering process. The smooth surface seen in microstructure of Figure 5a was obtained using 50% $O_2$ and a deposition power of 500 W, which caused small particles to pack tightly together. Microstructure observed in Figure 5b, on the other hand, was formed using the same oxygen percentage but with a higher deposition power of 1000 W, resulting in a more porous surface. The formation of growth lines and valleys with a cauliflower appearance, as seen in microstructure depicted in Figure 5c, occurred with a higher oxygen percentage of 75% and a power setting of 500 W. However, the microstructure, exhibited in Figure 5d, had growth lines with smoother valleys and mountain ranges of coalesced particles as the power setting was increased to 1000 W while keeping the oxygen percentage at 75%.

By analyzing Figure 5a, it is possible to verify that the conditions of a lower power setting of 500 W and an oxygen percentage of 50% used to deposit the film did not allow enough time for the particles to coalesce and form an uniform dense film. The lower power setting resulted in a smoother surface due to the lower kinetic energy of the particles [43]. This can be explained by considering the relationship between deposition rate and surface roughness in sputtering. The deposition rate is proportional to the sputtering power. The surface roughness, on the other hand, is influenced by the kinetic energy of the particles arriving at the surface. At lower kinetic energies, the particles have less energy to penetrate into the substrate and create deeper features. Additionally, the particles have a reduced ability to overcome surface energy barriers and form new bonds with the substrate. As

a result, the surface roughness is reduced, and, consequently, the surface morphology is smoother. On the other hand, Figure 5b shows a porous surface with larger gaps between particles due to a higher power setting of 1000 W and an oxygen percentage of 50%, which resulted in a more energetic deposition process and particles that are not as closely packed, hence the porous surface that is observed. The explanation for this can also be found by considering the relationship between deposition rate and surface roughness, as explained above. The observation of the formation of "growth lines" and valleys with a cauliflower appearance in Figure 5c can be explained by the fact that the film was created using a lower power setting of 500 W and a higher oxygen percentage of 75%. At higher oxygen percentages, the oxidation of the target material leads to the formation of oxide compounds on the surface of the film, which can cause the observed surface morphology. When the oxygen content in the atmosphere is high, the target material is more likely to oxidize, leading to the formation of oxide compounds on the surface of the film [44].

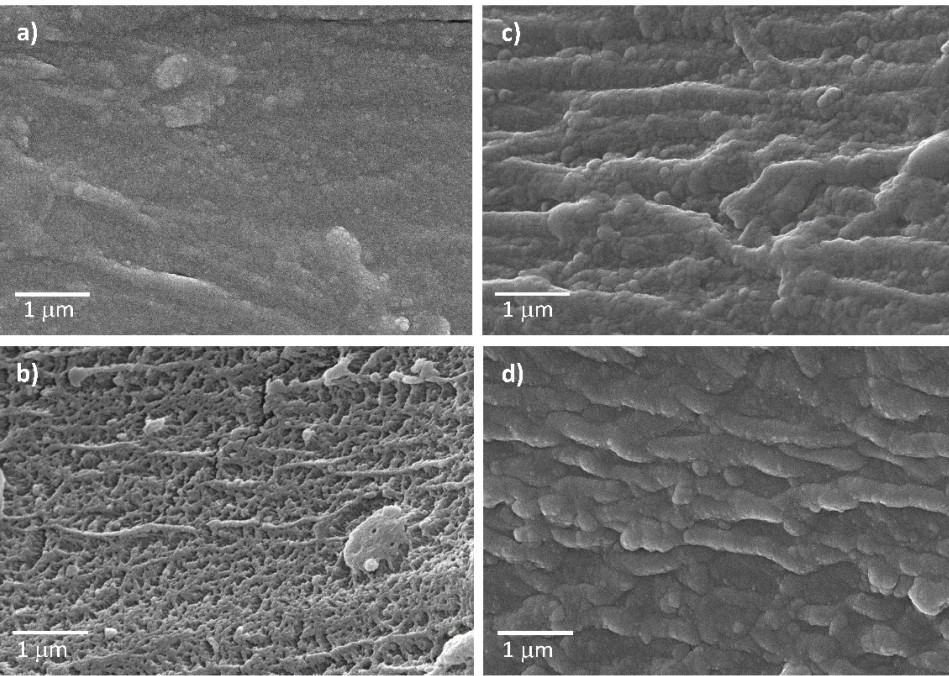

**Figure 5.** SEM surface images of $TiO_2$ thin films obtained by DC reactive sputtering under different conditions: (**a**) 50%$O_2$–500 W, (**b**)50%$O_2$–1000 W, (**c**)75%$O_2$–500 W, (**d**)75%$O_2$–1000 W.

As the film grows, the high points on the surface of the fiber receive more deposited material due to their increased surface area. This can result in the formation of columnar structures where the high points coalesce and grow in height. The deposited particles at the valleys or lower regions between the high points receive less material and may remain as voids or gaps in the film. As the high points grow in height, they can recover and coalesce with neighboring high points, resulting in the formation of "growth lines" in the microstructure. Growth lines are visible lines or ridges that can be observed on the surface of the film and are indicative of the columnar growth mode. The resulting microstructure can be described as columnar or dendritic, where the high points form the branches or columns and the valleys or gaps form the spaces between the columns.

The presence of growth lines with smooth valleys detected in Figure 5d is due to the higher power setting of 1000 W and a higher oxygen percentage of 75% used during film creation. The increased power setting resulted in a more energetic deposition process, which means that more energy was available to the particles during the sputtering process. This, in turn, led to the formation of larger particles on the fiber that can lead to the development of growth lines with smooth valleys observed on the fiber's surface.

In order to evaluate the 2D surface roughness parameters, the SEM images of Figure 5 were analyzed by the Mountains®9 software and results are shown in Figure 6. The software follows the ISO 21920—Roughness (S–L) for 2D surface roughness calculation [45] and the following parameters were used for quantifying average surface texture [46]:

$$Ra = \frac{1}{L}\int_0^L |z(x)|dx \qquad (4)$$

where $Ra$ is the arithmetic mean roughness, $L$ is the evaluation length and $z(x)$ is the surface profile height deviation from the mean line at position $x$.

$$Rq = \sqrt{\frac{1}{L}\int_0^L z^2(x)dx} \qquad (5)$$

where $Rq$ is the root mean square roughness, $L$ is the evaluation length, and $z(x)$ is the surface profile height deviation from the mean line at position $x$.

$$Rz = \frac{1}{5}[(Z1 - V1) + (Z2 - V2) + \ldots + (Z5 - V5)] \qquad (6)$$

where $Rz$ is the ten-point height roughness, which is the average distance between the five highest peaks and the five lowest valleys within a specified evaluation length. $Z1$ to $Z5$ are the heights of the five highest peaks and $V1$ to $V5$ are the depths of the five lowest valleys within the evaluation length.

$$Rsk = \frac{1}{N}\sum \frac{(z_i - z_m)^3}{\sigma_z^3} \qquad (7)$$

where $Rsk$ is the skewness which is a measure of the asymmetry of the surface texture profile about the mean line. $N$ is the number of data points in the profile, $z_i$ is the $i^{th}$ height deviation from the mean line, $z_m$ is the mean height deviation, and $\sigma_z$ is the standard deviation of the height deviations.

$$Rku = \frac{1}{N}\sum \frac{(z_i - z_m)^4}{\sigma_z^4} - 3 \qquad (8)$$

where $Rku$ is the kurtosis, which is a measure of the sharpness of the peaks and valleys in the surface texture profile. $N$ is the number of data points in the profile, $z_i$ is the $i^{th}$ height deviation from the mean line, $z_m$ is the mean height deviation, and $\sigma_z$ is the standard deviation of the height deviations. The $Rku$ equation contains the term "$-3$" because it is a correction for the effect of kurtosis on the normality of the distribution. A normal distribution has a kurtosis of 3, so if the calculated kurtosis is 3, then the distribution has the same shape as a normal distribution. If the calculated kurtosis is greater than 3, then the distribution has a sharper shape, with a higher concentration of values near the peaks and valleys. If the calculated kurtosis is less than 3, then the distribution has a flatter shape, with values closer to the mean line and a fewer values near the peaks and valleys. The subtraction of 3 in the $Rku$ equation is used to normalize the kurtosis to facilitate comparison between different distributions.

ISO 21920 defines roughness parameters for characterizing the surface roughness of materials, typically in the field of metrology and surface texture measurement. The "(S–L)" in the title refers to the use of S-filter and L-filter in the measurement process, which are specific types of Gaussian filters used for filtering surface data in order to calculate the roughness parameters. The obtained roughness parameters $Ra$, $Rq$, $Rz$, $Rsk,$ and $Rku$ are summarized in Table 2. These parameters can provide insight into the average height, the root-mean-square roughness, the maximum peak height, and the maximum valley depth of the surface features, respectively.

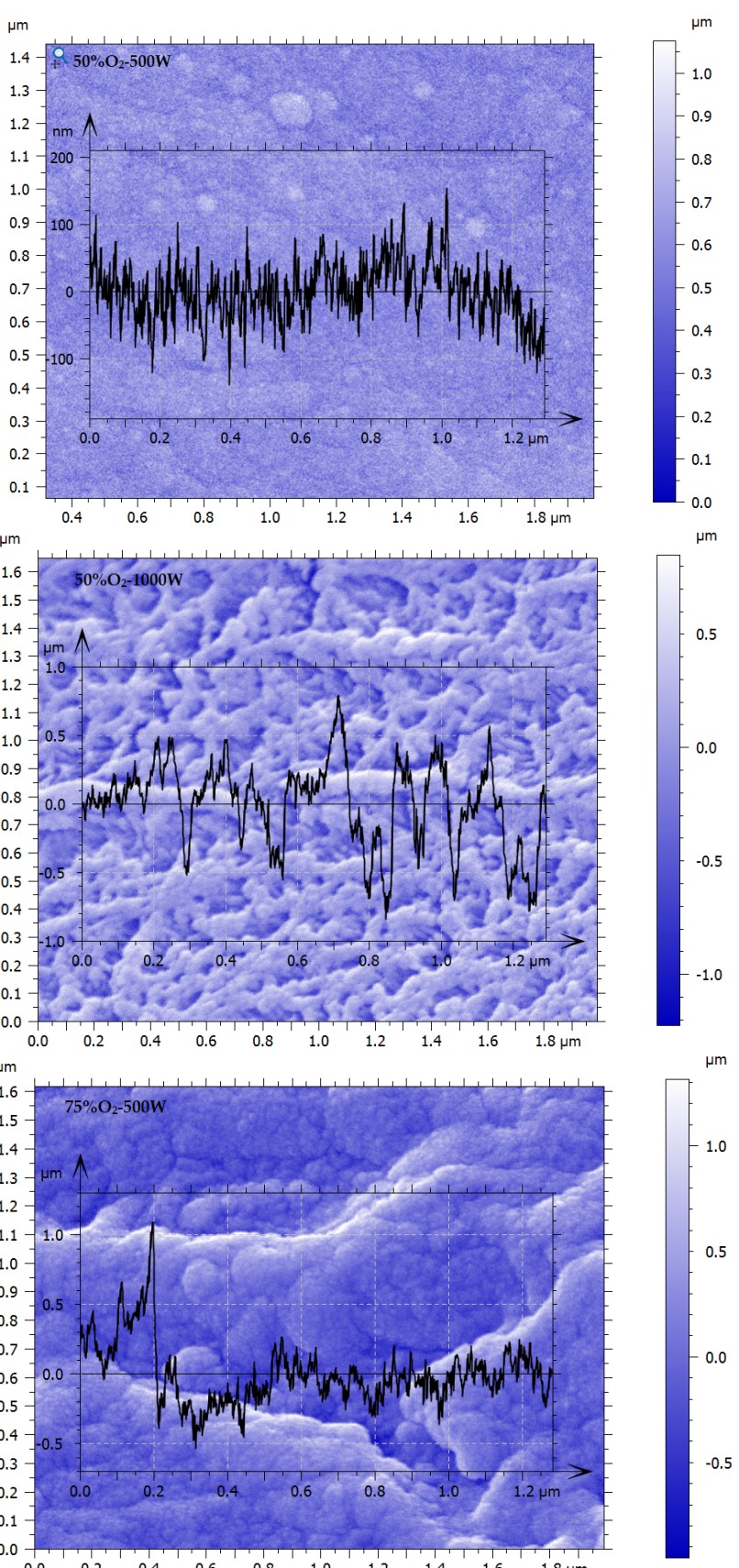

**Figure 6.** *Cont.*

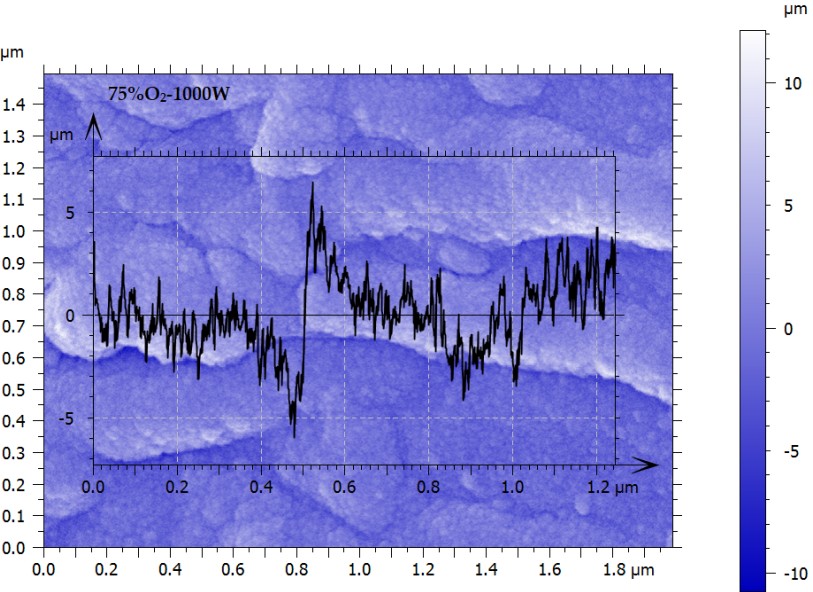

**Figure 6.** Roughness profiles of the SEM surface images of TiO$_2$ thin films obtained by DC reactive sputtering under different conditions: 50%O$_2$–500 W, 50%O$_2$–1000 W, 75%O$_2$–500 W, and 75%O$_2$–1000 W.

**Table 2.** The roughness parameters *Ra, Rq, Rz, Rsk,* and *Rku* of the sputtered TiO$_2$ films at 500 and 1000 W and deposited with 50% and 75% O$_2$.

| Sample | ISO 21920—Roughness Parameters (S–L) | | | | |
|---|---|---|---|---|---|
| Film | Rq (nm) | Rsk | Rku | Rz (nm) | Ra (nm) |
| 50%O$_2$–500 W (*) | 5.009 | 0.2396 | 2.189 | 13.58 | 4.145 |
| 50%O$_2$–1000 W | 67.50 | 0.3078 | 2.123 | 245.3 | 56.36 |
| 75%O$_2$–500 W | 77.80 | −0.2808 | 1.895 | 279.0 | 68.10 |
| 75%O$_2$–1000 W | 467.9 | −0.1675 | 2.084 | 1741 | 393.7 |

Note: The reported values were obtained using an S-filter (Gaussian filter with a cut-off wavelength of 0.0008 mm) and an L-filter (Gaussian filter with a cut-off wavelength of 0.0025 mm), except for the measurement indicated with an asterisk (*), which used a different cut-off wavelength for the L-filter of 0.0008 mm. *Rsk* and *Rku* are dimensionless parameters.

Upon analyzing Table 2, a few observations can be made:

1. The surface roughness parameters vary significantly between the different film samples, indicating that the deposition conditions have a significant impact on the resulting surface topography;

2. Increasing the oxygen concentration in the deposition atmosphere from 50% to 75% appears to increase the surface roughness parameters for both power levels, as seen by the increase in *Rq*, *Rz*, and *Ra* for 75% O$_2$ compared to 50% O$_2$. Indeed, comparing the 50% O$_2$ films at 500 and 1000 W, we can see that Rq increases significantly from 5.009 nm to 67.50 nm, indicating an increase in surface roughness with increasing the sputtering power level. Comparing the 500 W films at 50% and 75% O$_2$, it can be detected that *Rq* increases from 5.009 to 77.80 nm, indicating an increase in surface roughness with increasing oxygen concentration;

3. Increasing the sputtering power from 500 to 1000 W also increases *Rz*, and *Ra*. Comparing the 75% O$_2$ films at 500 and 1000 W, it is observed that Rz increases from 279.0 to 1741 nm, indicating an increase in the maximum peak-to-valley height of the surface with increasing power level. Additionally, comparing the 75% O$_2$ films at 500 and 1000 W, *Ra* increases from 68.10 to 393.7 nm, indicating an increase in the average roughness of the surface with increasing sputtering power level. This indicates that the surface becomes rougher on average with the rise of sputtering power level.

4.  The *Rsk* and *Rku* parameters show some variation between the different films, indicating that the surface height distribution is not perfectly symmetric and has some degree of deviation from a Gaussian distribution. For example, comparing the 50% $O_2$ and 75% $O_2$ films at 500 W, it is observed that *Rsk* decreases from 0.2396 to $-0.2808$, indicating a shift towards a more symmetric surface height distribution with increasing oxygen.

### 3.6. TiO₂ Thin Film-Fiber Adherence

To determine the adhesion of the $TiO_2$ film to the fiber, the quantity of Ti was assessed using energy-dispersive X-ray spectroscopy (EDS) before and after subjecting it to sonication in a water bath. Sonication is a process that involves exposing a sample to high-frequency sound waves, and it is commonly used to evaluate the strength of a thin film's bond with a substrate. In this study, the sonication was performed continuously at 70% amplitude.

After sonication, any $TiO_2$ film regions that were weakly adhered to the fiber were assessed for detachment, and the integrity of the coatings was evaluated based on the percentage of Ti loss. The detachment was rated on a linear scale from 0 (indicating no remaining Ti) to 5 (indicating Ti fully remaining), with a corresponding percentage of Ti remaining on the sample. For instance, a grade of 2.5 corresponded to approximately 50% Ti remaining. The equation used to calculate the percentage of TiO2 detachment was

$$\%\text{Ti} = (\text{Ti\_before} - \text{Ti\_after})/\text{Ti\_before} \times 100 \tag{9}$$

where %Ti represents the percentage of $TiO_2$ detachment from the substrate, Ti_before is the initial $TiO_2$ content before sonication, and Ti_after is the $TiO_2$ content after sonication. The EDX data analysis revealed that some $TiO_2$ had detached from the fiber after sonication. The units of %Ti obtained from EDS were in atomic percent (at. %). The difference between the amounts of $TiO_2$ before and after sonication was indicative of the degree of detachment. A lower percentage of detachment indicated the better adhesion of the film to the substrate, while a higher percentage of detachment indicated poor adhesion or bonding between the film and the fiber. Table 3 summarizes the grades (out of 5) achieved after the sonication process of the sputtered $TiO_2$ films under different conditions (50%$O_2$–500 W, 50%$O_2$–1000 W, 75%$O_2$–500 W, and 75%$O_2$–1000 W). The remaining Ti (at. %) and Ti (at. %) loss for each film grade are also provided, ranging from 86% to 96% and 4% to 14%, respectively. The grades are rated on a scale of 1 to 5, with 5 being the highest grade. Almost all films were graded around 4.7, i.e., with Ti almost fully remained after sonication, except for the film that was porous, which exhibited a lower grade of 4.3. Therefore, the adhesion is generally good for the sputtered $TiO_2$ films on these plant fibers surfaces.

**Table 3.** Grades after sonication process of the sputtered $TiO_2$ films at 500 and 1000 W and deposited with 50% and 75% $O_2$.

| Film | Grade (Out of 5) | Ti (at. %) Remaining | Ti (at. %) Loss |
|---|---|---|---|
| 50%$O_2$–500 W | 4.7 | 94 | 6 |
| 50%$O_2$–1000 W | 4.3 | 86 | 14 |
| 75%$O_2$–500 W | 4.8 | 96 | 4 |
| 75%$O_2$–1000 W | 4.8 | 96 | 4 |

In fact, in an oxygen-rich environment, the Ti atoms react with the oxygen molecules in the plasma, leading to the formation of Ti–O bonds. The more oxygen there is in the plasma, the more likely it is that Ti atoms will react with them to form Ti–O bonds. As a result, increasing the partial pressure of oxygen during sputtering can lead to the formation of more tightly packed Ti–O bonds and a higher packing density of the resulting $TiO_2$ film.

This higher packing density can have several effects on the properties of the deposited $TiO_2$ film. For example, it can improve the adhesion of the film to the substrate and make it more resistant to cracking and peeling.

## 4. Conclusions

The study examined the effect of sputtering power and oxygen concentration on the crystalline structure and surface morphology of $TiO_2$ films deposited on ginger lily fibers through Reactive DC Magnetron Sputtering. The XRD analysis revealed that the films produced at 500 W were amorphous, while those produced at 1000 W presented a polycrystalline structure consistent with $TiO_2$ anatase. The dominant phase of the films produced with 50% $O_2$ was anatase, while those produced with 75% $O_2$ exhibited a significantly lower presence of rutile phase in addition to anatase.

The SEM microstructures of the films were also affected by the deposition conditions. When a deposition power of 500 W and 50% $O_2$ were used, a smooth surface was obtained. On the other hand, a more porous surface was produced with a higher deposition power of 1000 W. When the oxygen percentage was increased to 75% and the power setting was kept at 500 W, growth lines and valleys with a cauliflower appearance were observed. However, increasing the power setting to 1000 W while maintaining the oxygen percentage at 75% resulted in coalescent growth lines with smoother valleys.

The surface roughness parameters of film samples are also significantly affected by the same deposition conditions. Increasing the oxygen concentration generally increases the surface roughness, while increasing the sputtering power level can result in an increase in the maximum peak-to-valley height and an increase in the average roughness. Additionally, the surface height distribution is not perfectly symmetric and can deviate from a Gaussian distribution.

The study's findings of good adhesion of $TiO_2$ films to ginger lily fibers suggest the potential use of these films in various applications. Furthermore, the study highlights that the final crystalline structure of the films is primarily affected by the sputtering power used during the deposition process, whereas the surface morphology is influenced by a tailored combination of deposition parameters, such as sputtering power and gas mixture. This valuable insight can guide the optimization of deposition conditions for $TiO_2$ films on plant fiber substrates, facilitating the development of novel and improved films with customized properties for specific sustainable and environmentally friendly applications.

**Author Contributions:** Conceptualization, H.C.V.; methodology, H.C.V. and T.E.; software, T.E.; formal analysis, S.S. and T.E.; investigation, S.S. and T.E.; data curation, H.C.V.; writing—original draft preparation, all authors; writing—review and editing, all authors; supervision, H.C.V.; funding acquisition, all authors. All authors have read and agreed to the published version of the manuscript.

**Funding:** This research was funded by (1) FEDER, through Programa Operacional Factores de Competitividade (COMPETE) and Fundação para a Ciência e a Tecnologia (FCT) by the project No. UIDP/FIS/04559/2020 (LIBPhys); (2) FEDER, through POACORES-Valorização e Desenvolvimento de Produtos da Conteira (*Hedychium gardnerianum*)-01-0247-FEDER-000011; (3) Regional Government of the Azores-Fundo Regional da Ciência e Tecnologia (Fellowship M3.1.a/F/040/2015).

**Institutional Review Board Statement:** Not applicable.

**Informed Consent Statement:** Not applicable.

**Data Availability Statement:** Not applicable.

**Acknowledgments:** The authors wish to thank to Anne Berger, Sales Manager, Digital Surf, Besancon, France for permission to use MountainsMap® 9 version software P(Trial version) (Digital Surf, Besançon, French) for SEM image processing.

**Conflicts of Interest:** The authors declare no conflict of interest.

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
