# Peer review of "Growth of Nanostructured TiO2 Thin Films onto Lignocellulosic Fibers through Reactive DC Magnetron Sputtering: A XRD and SEM Study"

_coatings, doi:10.3390/coatings13050922_

Round 1

Reviewer 1 Report (Previous Reviewer 3)

I think that the manuscript is good for publication.

Reviewer 2 Report (Previous Reviewer 2)

As the Reviewer understands, this manuscript has been re-submitted after the original version of the paper was rejected. In the new version, all the comments by the Reviewer in regards to the flaws of the original version have been considered. The Reviewer belives that the paper can now be accepted as is.

Reviewer 3 Report (Previous Reviewer 1)

All questiones have been issued,  and I think it can be accepted.

The English language is acceptable.

This manuscript is a resubmission of an earlier submission. The following is a list of the peer review reports and author responses from that submission.

Round 1

Reviewer 1 Report

In this manuscript, TiO2 thin films were deposited on ginger lily fibers using a custom-made DC reactive magnetron sputtering system with Ar/O2 mixture and talk about the phase and morphology using XRD and SEM. However, a previous paper by the author had been published to show the preparation method together with XPS and FTIR characterization. In this manuscript, the preparation condition is the same with the previous one, and only show the XRD and SEM results. There are no new scientific discovery in this manuscript. Before submitting similar research about this topic, the author need to find at least one application of this composite materials and test its performance. Here are some comments which should be carefully issued.

1. In figure 4, standard peaks of anatase and rutile phase should be given to compare with the obtained results.

2. SEM images with large magnification should be given to check the morphology of TiO2. Besides, TEM images are also needed.

3. The author should find at least one application of this composite materials and test its performance.

Reviewer 2 Report

The paper is devoted to the XRD and SEM study of TiO2 thin films deposited onto lignocellulosic fibers through reactive DC magnetron sputtering under various conditions (power value and plasma composition). The subject of the study seems interesting, the paper in general is well written, and the conclusions are supported by the data. The only comment (in addition to the neccessity of misprint double-check) relates to some unneccessary and excessive, in the Reviewer's opinion, information presented in the paper. This relates to sections 2 and 3 (Fundamentals of reactive DC magnetron sputtering: a theoretical overview, etc), discussion of physical and chemical parameters that can influence the deposition of films by DC reactive sputtering in section 5.3, and reactions during the sputtering in section 5.5.  The Reviewer believes that these sections can be omitted or drastically shortened, which would make the paper easier to read and will increase its value.

Reviewer 3 Report

The manuscript deals with synthesizing TiO2 film on the Lignocellulosic Fibers showing the microstructure and morphology of the film growth. The comments are attached here.

Reviewer 4 Report

The title can be improved by removing the mention of "XRD and SEM Study," since these techniques are commonly used in such research.

Regarding the abstract, the motivation and objectives are not clearly stated. I recommend the authors restructure the abstract to address the following points:

1.       Briefly mention the importance of TiO2 thin films and their potential applications, especially when coated onto lignocellulosic fibers.

2.       Clearly state the purpose of the study.

3.       Concisely present the main findings, focusing on the differences observed in structure and surface morphology, as well as adhesion properties, under various deposition conditions.

Sections such as "2. Fundamentals of Reactive DC Magnetron Sputtering: Theoretical Overview" and "3. Common Parameters Used for Quantifying Average Surface Texture" might be redundant in an original research article, as the readers of Coatings likely already have a background in the relevant concepts and techniques.

Instead, the authors should focus on presenting their own original research and findings, while briefly referencing or citing well-established sources to support their methodology, if necessary. There is no need in trying to condense what was already written in various textbooks.

Additionally, it would be helpful if the authors addressed the following concerns in the methodology and results sections:

Provide more information about the selection of the deposition parameters, such as the rationale behind choosing specific O2/(O2+Ar) ratios and sputtering powers.

Elaborate on the test methodologies, such as the type of geometry of XRD measurement being used, how the samples for the XRD study were prepared, clearly you cannot get a proper signal from a single fiber. For example, what is the rationale for 2θ angles for fibers? It is hard to assess the profile of the XRD where only 2 broad peaks are represented. A deep analysis of XRD results is not present. It would be more useful and representative to perform all measurements in 2θ angles ranging from 10 to 50°. The adhesion test based on sonication may not be well-established, and the set of experiments and their representation are questionable.

Figure 2 must be reworked. It is clear that the XRD profiles “after deposition” are simulated and not original XRD patterns. Comparing treated and untreated samples in such a narrow 2θ angles window is also questionable.

In subsection 5.3, the redundant description of DC magnetron sputtering process parameters is listed again. There is no need for this, as it has already been described in various handbooks.

Lines 427 to 446 should be moved to the methodology section and described in more detail. For example, which peaks were analyzed using the Williamson-Hall method, and why? How big is the error of identification when a substantial part of the material is amorphous? Is the "average crystallite size" the actual size of crystallites, or is it a coherent scattering region and the actual crystallites tend to be smaller? How can this statement be proven?

Roughness measurement using only plain SEM images may result in ambiguity. For roughness measurement, it is advisable to perform stereographic imaging using various angles.

In conclusion, although the topic might be of interest, the methodology and representation of experimental results are questionable. Furthermore, the amount of obtained and analyzed research data may not be sufficient for publication in such a reputable journal. It seems that the authors tried to artificially increase the amount of text by adding information widely known to the professional community. I would be glad if I am wrong. As of now, I suggest rejecting the manuscript, significantly reworking it, and adding new methods of analysis.